# Netball Shoots for Physical and Mental Wellbeing in Samoa: A Natural Experiment

**DOI:** 10.3390/ijerph19052663

**Published:** 2022-02-25

**Authors:** Justin Richards, Emma Sherry, Fetuao Tamala, Suzie Schuster, Nico Schulenkorf, Lewis Keane

**Affiliations:** 1Faculty of Health, Victoria University Wellington, Kelburn, Wellington 6012, New Zealand; 2Department of Management & Marketing, Swinburne University of Technology, Melbourne, VIC 3122, Australia; esherry@swin.edu.au; 3Netball Samoa, Taga, Savai’i, Samoa; ftamala@savaiisisifo.schoolnet.ws; 4Faculty of Education, National University of Samoa, Apia, Upolu, Samoa; ss666@students.waikato.ac.nz; 5Business School, University of Technology Sydney, Ultimo, NSW 2007, Australia; nico.schulenkorf@uts.edu.au; 6School of Public Health & Charles Perkins Centre, University of Sydney, Camperdown, NSW 2050, Australia; lkea3312@uni.sydney.edu.au

**Keywords:** sport, sport-for-development, physical activity, exercise, Pacific Islands, developing countries

## Abstract

Sport-for-development programs claim to address key determinants of recreational physical activity participation and subsequent development outcomes in low-income settings. We conducted a natural experiment with pre–post measures taken from women in the 12 villages in Samoa, some of which voluntarily participated in the sport-for-development intervention. The intervention comprised a six-week netball league delivered by local volunteers who attended coaching workshops, received ongoing support from the national governing body and were provided with infrastructure and equipment to conduct local training sessions. Changes in netball participation, recreational physical activity, body composition, mental wellbeing and socio-ecological determinants of physical activity were compared between intervention and comparison villages using a univariate ANOVA. The intervention reached women who participated in little recreational physical activity and had poor physical and mental wellbeing. Program uptake was higher in villages with the strongest social support for netball participation. Local social support and capacity to independently organize netball activities increased. There were concurrent improvements in netball participation, physical activity levels, mental wellbeing and body weight in the intervention villages. Our findings support scaling-up of the intervention in similar settings but preceding this with formative evaluation to identify low active communities that are “primed” to participate in the proposed activity.

## 1. Introduction

Physical inactivity is the fourth leading risk factor for global mortality and contributes to numerous cardiovascular and metabolic non-communicable diseases (NCDs) [1]. This is particularly the case in Samoa, where the majority of the population do not meet the physical activity recommendations (i.e., 51.7%) and the prevalence’s of diabetes (i.e., 22.1%) and overweight/obesity (i.e., 85.2%) are amongst the highest in the world [2,3,4]. Women and girls in Samoa are at a higher risk of NCDs than males, of which 69.9% are insufficiently active: 23.4% have diabetes and 89.8% are overweight or obese [3,4]. There is also established evidence that participating in quality physical activity can promote mental wellbeing [5]. This may be particularly critical in the prevention of premature deaths as a result of NCDs [6]. Despite efforts to address the mental wellbeing needs within the Pacific region, suicide rates remain higher than global averages and local data suggests mental disorders are common [7,8].

In a previous study, we found that the prevalence of poor physical and mental wellbeing was high for women in rural settings on the island of Savai’i in Samoa [9]. Specifically, 83.3% of the sample were overweight or obese and 79.2% were classified as “at risk” for their mental wellbeing [9]. This was accompanied by relatively low participation in recreational physical activity that was also strongly associated with poor mental wellbeing [9]. These results were part of a formative evaluation for the planning of the One Netball Pacific initiative in Samoa, which was a sport-for-development program that aimed to promote physical and mental wellbeing and was part of the Australian Government’s Pacific Sport Partnership [10]. There are positive signs for the potential success of such programs in Samoa, with over 50% of the 13–50 age group wanting to be more physically active and 56% wanting to lose weight [11]. There also appears to be a disconnect in the perception of physical activity levels, with many Samoans believing they are sufficiently active, but not meeting the WHO recommendations [11]. Despite the clear need for intervention in this population group and the rapid proliferation of sport-for-development programs internationally, there remains a paucity of evidence for the impact on physical and mental wellbeing outcomes [12,13,14]. Our interest in evaluating the impact of promoting netball in Samoa stems from clear evidence that the local population would like to participate more and data indicating its potential to positively change physical activity levels [10,15].

Previous evaluations of the delivery and impact of the One Netball Pacific initiatives in Samoa have not included any quantitative process or outcome indicators [10]. Therefore, the primary objective of this paper was to quantify the impact of a new program on the netball participation, recreational physical activity levels, mental well-being and body composition of women living in a rural setting in Samoa. The program was specifically designed to address several key determinants of netball participation for these women. This included improving access to opportunities and local community support for playing netball. Consequently, our secondary objective was to measure changes in these identified determinants of netball participation that resulted from the new initiative. In completing this evaluation, our aim was to inform future replication and scale-up of this program across Samoa. We also anticipated improving broader decision-making processes for program implementation amongst sport-for-development stakeholders by adding to the emerging evidence base for the sector.

## 2. Materials and Methods

### 2.1. Study Setting

The study was set on the island of Savai’i in Samoa, a middle-income country located in the Polynesian region of the Pacific. Despite being the geographically larger of the two main inhabited islands of Samoa, only just over 20% of the national population of approximately 195,000 people live on Savai’i. Most of the population in Savai’i still live in small coastal villages around the island and lead more traditional lifestyles in a rural setting (i.e., including subsistence fishing and gardening).

### 2.2. Study Design

The study was a natural experiment with pre–post measures taken from women in the 12 villages located closest to the intervention site in 2015. All of these villages were given the opportunity to engage in the intervention. We anticipated that not all villages would participate, but we did not know the rate of uptake at the time the pre-measures were taken.

### 2.3. Participants

Study participants were conveniently sampled by approaching local residents with street-surveys and door-knocking houses and small businesses in the target villages. The sample size was calculated using data from the formative evaluation of netball participation and standard formulas were used based on an established relevant difference (δ = 1.0) [16], standard deviation (σ = 1.70) [9], probability of a type I error (α = 0.05) and probability of a type II error (β = 0.10). We assumed that 50% of the villages measured at baseline would register for the intervention and allowed for a 15% loss to follow-up. Therefore, 72 women were required for each group (i.e., total *n* = 144) and we recruited 12 women in each of the 12 participating villages. The data collected from one woman was lost in the upload process at baseline.

A total of seven villages originally registered to participate in the intervention, but only four of these ended up taking part. The reasons cited were inadequate transport, insufficient equipment, poor internal leadership and lack of local support. All women in the participating villages were exposed to the intervention and the women measured in the eight remaining villages comprised the comparison group.

### 2.4. Intervention

The intervention consisted of a six-week netball league that took place in the most centrally located village. The league was a round robin tournament, with each team playing one organized game per week on Saturday. Each of the 12 assessed villages were invited to enter teams in the league at no cost. All of these villages were “primed”, being provided with at least three balls, two sets of playing uniforms, two goal posts and the option to send a volunteer to attend a basic coaching course organized by Netball Samoa during the previous 12 months. These volunteers formed a network of local “champions”, who led the promotion and development of netball in their communities. They were supported by a local employee of Netball Samoa who coordinated the delivery of the league. The local “champions” were also encouraged to organize training sessions during the week at their own discretion, and these appeared to consistently occur in the afternoon on each weekday. There was no limit on the number of players each village included in the intervention, and it was evident that more women in each village were participating in the regular training sessions than the central league games. All of the intervention components were funded by Netball Australia as part of the Australian Government’s Pacific Sports Partnership.

### 2.5. Measurement

The survey questions were adapted from a previously published protocol [17].

Data for several sociodemographic variables were collected. Age was self-reported in years and categorized into three groups: (1) 15–24 years; (2) 25–34 years; (3) ≥35 years. Educational attainment referred to the highest level completed and was categorized into three groups: (1) Primary; (2) Secondary; and (3) Tertiary. Self-reported employment status was categorized into four groups: (1) Employed (government, private sector, non-government organization); (2) Self-employed (self-employed, plantation, homemaker); (3) Unemployed (unemployed); and (4) Other (student, retired, non-paid).

We developed quantitative indicators for key modifiable determinants of netball participation based on observations and interviews with key local stakeholders, existing international literature and our experiences in similar settings [18,19,20]. Specifically, we developed questions to understand interpersonal determinants of netball participation that focused on the support from elders, community, family, friends and church (i.e., 5-point Likert scale). We also developed survey items to understand organisational determinants of participation that included competence of local netball personnel (i.e., visual analogue scale for confidence) and enjoyment of existing netball opportunities (i.e., 5-point Likert scale). These questions were translated into Samoan by a local translation team and back translated into English for quality control. They were then pre-tested for face validity with 15 Samoan participants from various demographics and further refined accordingly.

Regularity of netball participation was assessed based on the number days in a typical week the participant played netball (i.e., 1 day, 2 days… 7 days), with additional response options of “less than once per week” and “more than daily”. Physical activity participation was measured using the relevant sections of the Global Physical Activity Questionnaire (GPAQ), which is a validated tool that has been widely used in low- and middle-income settings [21]. Mental wellbeing was assessed using the WHO-5 index, which is a reliable and validated measure that is widely accepted internationally and has been used across various clinical and general population samples [16]. Body Mass Index (BMI) was calculated with objective measures of height and weight, which were assessed by applying standardized methods previously tested and used in similar settings (i.e., locally acquired scales and telescopic ruler) [22].

### 2.6. Data Collection

Two local data collectors were trained by the research team to conduct the survey and complete the objective body composition measures for all participants. The baseline measures were collected prior to when villages opted in or out of the intervention and the data collectors were blinded to whether a village participated when post-measures were taken. The contact details for each participant were recorded at baseline to facilitate tracking for follow-up measurement and no participants were lost to follow-up. All data were collected in a location chosen by the participant that was immediately accessible, private and comfortable. The data collectors used iPadsTM with iSurveyTM software installed, which uploaded all results to the cloud when access to a wireless network was available.

### 2.7. Data Analysis

The descriptive statistics for the sociodemographic factors, netball participation, recreational physical activity, body composition and mental wellbeing were calculated and tabulated for all participants and separately for each group (i.e., intervention vs. comparison). Netball participation was dichotomized according to weekly participation (i.e., days/week ≥ 1). The physical activity data was cleaned and processed according to established protocols for GPAQ [23]. The results for the total number of minutes of physical activity were then categorized to identify the proportion inactive (i.e., mins/week = 0) and according to the current physical activity recommendation (i.e., mins/week ≥ 150). BMI was categorized according to the international threshold defined for overweight (i.e., BMI ≥ 25) and obese (i.e., BMI ≥ 30). The results for the total WHO-5 score were dichotomized according to established thresholds for being “at risk” (i.e., total ≤ 12) [16].

The crude mean and standard deviation at baseline and follow-up for the measured determinants of netball participation and program outcomes were stratified according to intervention group. We tested between-group differences at baseline with a univariate ANOVA and within-group changes using a paired samples t-test. The between-group differences in mean change were compared using a univariate ANOVA and standardized effect sizes were also calculated using a pooled standard deviation. This was then repeated with adjustments for pre-specified covariates (baseline measures) and factors (baseline, age, village of residence, education level). All results were tabulated and the threshold for statistical significance was taken as *p* < 0.05.

### 2.8. Ethical Approval

This study was approved by the La Trobe University Human Research Ethics Committee (13-073) and the National University of Samoa. Approval was also granted by Netball Samoa and community leaders in the participating villages. All participants provided informed consent prior to commencing data collection.

## 3. Results

### 3.1. Sample Characteristics

Key characteristics of the overall sample and stratified by intervention group at baseline are presented in Table 1. The majority of the sample were aged 15–24 years, had completed education to a secondary school level and were either unemployed or students. The distribution of the intervention group appeared to be slightly younger and better educated with a higher proportion of participants that were currently students. At baseline, the majority of the samples were participating in netball less than once per week, completed very little other recreational physical activity, were overweight and had poor mental wellbeing. Women in the intervention group initially appeared to be participating in less netball and other recreational physical activity but had healthier body composition and mental wellbeing at the outset.

### 3.2. Intervention Reach and Impact

The summary statistics for the determinants of netball participation and program outcomes at baseline and follow-up are presented in Table 2. At baseline, support for netball participation was significantly higher in the intervention villages than the comparison villages from village elders (*p* = 0.004), local community (*p* = 0.016), family (*p* = 0.001), friends (*p* = 0.005) and the church (*p* = 0.003). Confidence in the local capacity to organize netball activities was also initially higher in the intervention villages (*p* < 0.001), but there was no significant difference in how much the women enjoyed the opportunities that were available (*p* = 0.235). There was no significant difference between the intervention and comparison groups for netball participation (*p* = 0.105), recreational physical activity (*p* = 0.222), body weight (*p* = 0.372) or mental wellbeing (*p* = 0.080) at baseline.

There was a significant improvement in the support provided for netball from village elders, local community, family, friends and the church in both the intervention and comparison groups (*p* < 0.001). Confidence in the local capacity to organize netball did not change in the intervention group (*p* = 0.172) but did significantly improve in the comparison group (*p* < 0.001). Enjoyment of locally available netball activities significantly improved in both the intervention and comparison groups (*p* < 0.001). There were also significant improvements in netball participation, recreational physical activity and mental wellbeing for both the intervention and comparison groups (*p* < 0.001). In contrast, there was no change in body weight for the intervention group (*p* = 0.101) and a statistically significant increase for the comparison group (*p* < 0.001).

The results from comparing the changes in the intervention vs. comparison groups are presented in Table 3. There was no significant difference in the changes that occurred in the support for netball from the local community, family, friends or the church when comparing intervention vs. comparison group. The difference in the crude results for support from village elders (*p* = 0.036) was no longer statistically significant (*p* = 0.220) when adjusted for baseline, age, village and education. Although the crude results indicated that the comparison group improved more in their confidence in local capacity to organize netball (*p* = 0.006), the adjusted analysis showed the contrary (*p* = 0.002) with a large, standardized effect size favouring the intervention group. The crude change in enjoyment of existing netball opportunities was not different between the groups (*p* = 0.168) but was significantly larger for the intervention group in the adjusted analysis (*p* < 0.001) and had a moderate standardized effect size. Crude changes in netball participation (*p* < 0.001), recreational physical activity (*p* < 0.001), body weight (*p* < 0.001) and mental wellbeing (*p* = 0.037) all favoured the intervention group. These were maintained in the adjusted analysis for netball participation (*p* < 0.001), recreational physical activity (*p* < 0.001), body weight (*p* = 0.002) and mental wellbeing (*p* < 0.001) with very large, standardized effect sizes.

## 4. Discussion

This study is a rare example of a comprehensive quantitative impact evaluation of a sport-for-development initiative. We found that participation in recreational physical activity was very low in rural parts of Samoa, and this was accompanied by poor physical and mental wellbeing. The villages most likely to register for a voluntary sports league were those that had the strongest support across key local stakeholders, but they were not different to other villages in terms of participation in recreational physical activity or physical and mental wellbeing outcomes at baseline. The six-week netball league appeared to have a positive impact that extended beyond the villages directly exposed to the intervention, particularly on local support for netball and the capacity to organize enjoyable netball opportunities. Although the changes in support for netball were consistent across the community, the villages registered for the intervention experienced greater benefits in the organization and enjoyment of netball activities. This was accompanied by concurrent community-wide improvements in netball participation, recreational physical activity levels and mental wellbeing that were also larger in the intervention villages. The netball league also appeared to positively effect body weight when compared to those not directly exposed to the intervention.

### 4.1. Program Reach

Our results indicate that the sport-for-development initiative was being delivered in a community where there was a genuine need for intervention. Specifically, the program had objectives to promote physical and mental wellbeing by increasing netball participation and there was a clear deficit in all of these outcomes at baseline. This aligns with the formative evaluation we previously completed with a similar sample of women in Samoa [9].

Our findings also indicated that the program reached those who were living in villages with the strongest local support for netball participation and capacity to organize netball activities. This suggests that the support provided to the villages in the 12 months prior to the netball league did not adequately “prime” all of them to partake in the program. It is possible that more intensive or alternative approaches may be needed to address barriers to netball participation prior to setting up future interventions comprising competitive leagues. Although these findings may be expected in a natural experiment and are consistent with a strengths-based approach to intervention delivery, the limited reach of voluntary physical activity programs to those most in need is often overlooked [24]. Specifically, sport-for-development programs with voluntary registration processes often primarily reach the population groups that are already the most active [24], but that does not appear to be the case in this evaluation.

Despite the differences in interpersonal and organisational physical activity determinants at baseline, the low levels of netball and physical activity across all villages in our sample suggests that other factors were the overriding determinants for participation. It is possible that intrapersonal (e.g., knowledge, skills, confidence) and/or environmental (e.g., access, transport, infrastructure) factors were critically deficient in all villages included in our study [18,19]. This is consistent with previous qualitative work that has focused on barriers to physical activity participation for women in Samoa [11,25,26]. Similarly, a study conducted in Tonga, which is a neighbouring Pacific Island Nation, identified several intrapersonal (i.e., body image, competition, clothing) and physical environment (i.e., travel time, equipment) mediators for recreational physical activity participation [20]. Future evaluations of similar interventions should include measures across the socio-ecological model to ensure all critical levers for intervention are identified.

### 4.2. Program Impact

The community-wide improvement in the interpersonal determinants of netball participation is most likely due to contamination of intervention effects into the comparison villages. The strong social, familial and religious links between villages supports anecdotal evidence that the program components aimed at increasing local support for netball were widely experienced within the intervention villages and also spread to other neighbouring villages. This aligns with previous evidence from the sport-for-development sector that has described improvements in a control group from intervention contamination and a local atmosphere of anticipation and excitement about a program [22]. An alternative explanation could be measurement error (e.g., social-desirability bias), but the locally developed survey items were tested for face validity and previous studies using these were sensitive to expected differences in similar population groups [9,17,20]. Despite the within-group improvements for the interpersonal determinants, the absence of any between-group differences in these changes suggests that other factors are critical for the effective promotion of netball and physical activity participation.

In contrast, the difference between the groups in the changes observed for each of the organisational determinants was aligned with that observed for netball and physical activity participation. Specifically, it appears that the program positively influenced the local capacity to organize enjoyable opportunities to play netball in the intervention villages more than in the comparison villages, and this corresponded with the changes in physical activity behaviour. This finding aligns with the program focus on developing local volunteers as “champions” of netball promotion. The cascade model of sport-for-development intervention delivery through local “champions” has been widely disseminated and the importance of these volunteers to the success of the program has been previously highlighted [27,28]. Our results further support the inclusion of rigorous capability building of carefully selected local “champions” to become key change agents in their communities. It is likely that they also positively influence the intrapersonal and environmental factors known to be important for physical activity behaviour change [18,19], but not reported in this study.

The changes in all of the program outcomes in favour of the intervention group provide strong evidence of the effectiveness of the intervention. It appears that the changes in netball participation largely align with the changes in recreational physical activity levels for both groups. This concurs with previous research suggesting that there are limited opportunities for recreational physical activity for women in rural settings of Samoa [25,26]. The increase in physical activity for the intervention group was to an average duration that was almost double the current global recommendations of 150 min per week [1]. This was accompanied by relative improvements in both physical and mental wellbeing that were commensurate with the magnitude and the nature of the change in physical activity participation [5,29]. Specifically, it appears that the provision of large volumes of enjoyable physical activity was sufficient to significantly improve the mental wellbeing and reduce the body weight of women in the participating villages.

### 4.3. Strengths and Limitations

In comparison to previous sport-for-development evaluations, the strengths of our study are the inclusion of a comparison group, sample size calculation, formative evaluation and quantitative measures for key determinants directly targeted by the program. However, there were several limitations to our study that compromise the strength of the conclusions that we can draw from the results. Specifically, the comparison group was not a true control as would be the case in an experimental study, because it was not randomly selected or matched and there were differences between the groups for key determinants of physical activity at baseline. These differences are likely to have influenced the engagement of each village in the intervention at the outset (i.e., determined whether or not they participated in the competitive netball league). However, this is a known limitation that can occur in natural experiments and differences at baseline were controlled for in the adjusted analysis [30]. It is also important to note that there were no differences between the groups in the program outcomes at baseline. The uneven group sizes and relatively small number of people exposed to the intervention also appeared to be a limitation of the study. However, the changes in the variables measured exceeded expectations and the magnitude of the standardized effect sizes indicate that the study had adequate power. The positive changes seen in the comparison villages suggests contamination may have occurred across groups, which is a phenomenon that is difficult to control in evaluations of field programs in locations where people are mobile and interact socially [22,31]. However, the relatively large improvements seen in the intervention group and the concurrent changes in the mediating determinants indicate that the intervention villages received critical additional program delivery components and/or dosage. Ideally, we would have collected data on broader range of determinants as indicated in the formative evaluation and this should be a focus of future studies [9]. The sustained impact of the intervention should also be considered in follow-up studies with repeat measures taken at least 3–6 months after the completion of the intervention.

## 5. Conclusions

We conclude that the One Netball Pacific program and similar initiatives have the potential to positively influence physical activity participation and improve the physical and mental wellbeing of women in Savai’i. There is sufficient evidence to support replicating and evaluating the program in other parts of Samoa and similar settings globally. The voluntary nature of these interventions and their typical reach suggests that the greatest benefit is likely to be realized if they are targeted at villages where people are least active but are “primed” to participate. This may require alternative “priming” approaches that are informed by adequate formative evaluation of the determinants of recreational physical activity as part of planning and scaling up locally adaptable programs. Future initiatives should include broader measures across the entire socio-ecological model to understand what mediators of physical activity need to be addressed and how these are subsequently impacted by the program.

## Figures and Tables

**Table 1 ijerph-19-02663-t001:** Descriptive statistics for sociodemographic factors, netball participation, recreational physical activity, body composition and mental wellbeing for women at baseline.

	Total (*n* = 143)	Intervention (*n* = 47)	Comparison (*n* = 96)
**Age, mean (SD)**			
Years	23.10 (8.13)	21.36 (6.80)	23.96 (8.62)
**Age Group, *n* (%)**			
15–24 years	98 (68.5%)	34 (72.3%)	64 (66.7%)
25–34 years	28 (19.6%)	11 (23.4%)	17 (17.7%)
≥35 years	17 (11.9%)	2 (4.3%)	15 (15.6%)
**Education, *n* (%)**			
Primary	3 (2.1%)	1 (2.1%)	2 (2.1%)
Secondary	111 (77.6%)	35 (74.5%)	76 (79.2%)
Tertiary	29 (20.3%)	11 (23.4%)	18 (18.8%)
**Occupation, *n* (%)**			
Employed	13 (9.1%)	3 (6.4%)	10 (10.4%)
Self-employed	18 (12.6%)	4 (8.5%)	14 (14.6%)
Unemployed	43 (30.1%)	10 (21.3%)	33 (34.4%)
Other	69 (48.3%)	30 (63.8%)	39 (40.6%)
**Netball Participation, *n* (%)**			
<1 day/week	122 (85.3%)	43 (91.5%)	79 (82.3%)
**Recreational Physical Activity, *n* (%)**			
Inactive (0 min/week)	62 (43.4%)	10 (21.3%)	52 (54.2%)
Insufficient active (<150 min/week)	138 (96.5%)	44 (93.6%)	94 (97.9%)
**Body Composition, *n* (%)**			
Overweight or Obese (BMI ≥ 25)	119 (83.2%)	37 (78.7%)	82 (85.4%)
Obese (BMI ≥ 30)	59 (41.3%)	15 (31.9%)	44 (45.8%)
**Mental Wellbeing, *n* (%)**			
“At Risk” (≤12)	114 (79.7%)	30 (63.8%)	84 (87.5%)

**Table 2 ijerph-19-02663-t002:** Baseline and follow-up results for the determinants of netball participation and program outcomes.

	Intervention (*n* = 47)		Comparison (*n* = 96)	
	Baseline (SD)	Follow-Up (SD)	Baseline (SD)	Follow-Up (SD)
**INTERPERSONAL DETERMINANTS**				
Support of village elders (score/5)	4.04 (0.59) *	4.62 (0.49) ^#^	3.75 (0.50) *	4.58 (0.52) ^#^
Support of local community (score/5)	4.11 (0.37) *	4.64 (0.49) ^#^	3.95 (0.34) *	4.55 (0.52) ^#^
Support of family (score/5)	4.19 (0.40) *	4.74 (0.44) ^#^	3.95 (0.34) *	4.58 (0.52) ^#^
Support of friends (score/5)	4.13 (0.34) *	4.74 (0.44) ^#^	3.97 (0.23) *	4.60 (0.49) ^#^
Support of church (score/5)	4.09 (0.28) *	4.74 (0.44) ^#^	3.93 (0.30) *	4.57 (0.50) ^#^
**ORGANISATIONAL DETERMINANTS**				
Confidence in local capacity to organise netball (score/100)	50.77 (27.21) *	54.36 (28.39)	30.81 (11.17) *	48.39 (32.28) ^#^
Enjoyment of current community netball activities (score/5)	4.06 (0.32)	5.0 (0.0) ^#^	3.99 (0.40)	4.80 (0.40) ^#^
**PROGRAM OUTCOMES**				
Netball participation (days/week)	0.23 (0.98)	4.98 (0.15) ^#^	0.56 (1.38)	2.53 (2.16) ^#^
Recreational MVPA (min/week)	40.32 (57.54)	296.17 (58.40) ^#^	28.28 (48.99)	95.57 (132.23) ^#^
Body weight (kg)	78.66 (25.02)	77.91 (24.88)	82.55 (22.91)	84.03 (23.58) ^#^
Mental wellbeing (score/100)	44.00 (27.55)	91.57 (12.95) ^#^	36.17 (17.60)	72.25 (27.61) ^#^

* Statistically significant between-group difference at baseline (*p* < 0.05); ^#^ Statistically significant within-group change between baseline and follow-up (*p* < 0.05).

**Table 3 ijerph-19-02663-t003:** Difference between changes in the intervention and comparison groups for the determinants of netball participation and program outcomes.

	Crude		Adjusted ^a^	
	Difference (95%CI)	Effect Size	Difference (95%CI)	Effect Size
**INTERPERSONAL DETERMINANTS**				
Support of village elders (score/5)	−0.26 (−0.50 to −0.02) *	−0.38	0.12 (−0.08 to 0.33)	0.24
Support of local community (score/5)	−0.07 (−0.28 to 0.14)	−0.12	0.10 (−0.15 to 0.35)	0.18
Support of family (score/5)	−0.08 (−0.29 to 0.13)	−0.14	0.16 (−0.06 to 0.39)	0.30
Support of friends (score/5)	−0.02 (−0.20 to 0.17)	−0.03	0.12 (−0.11 to 0.35)	0.22
Support of church (score/5)	0.01 (−0.17 to 0.20)	0.03	0.15 (−0.06 to 0.36)	0.30
**ORGANISATIONAL DETERMINANTS**				
Confidence in local capacity to organise netball (score/100)	−13.98 (−23.82 to −4.14) *	−0.50	26.40 (10.22 to 42.59) *	0.88
Enjoyment of current community netball activities (score/5)	0.12 (−0.05 to 0.30)	0.25	0.18 (0.09 to 0.27) *	0.71
**PROGRAM OUTCOMES**				
Netball participation (days/week)	2.78 (2.00 to 3.55) *	1.26	2.33 (2.00 to 2.66) *	2.65
Recreational MVPA (mins/week)	188.56 (152.23 to 224.89) *	1.83	198.00 (174.17 to 222.16) *	3.04
Body weight (kg)	−2.22 (−3.23 to −1.21) *	−0.78	−2.25 (−3.62 to −0.88) *	−0.43
Mental wellbeing (score/100)	11.49 (0.70 to 22.28) *	0.38	16.50 (9.85 to 23.19) *	1.08

* Statistically significant between-group difference in mean change (*p* < 0.05); ^a^ Adjusted for baseline, age, village of residence, education level.

## Data Availability

The data presented in this study are available on request from the corresponding author.

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
