# Peer review of "Netball Shoots for Physical and Mental Wellbeing in Samoa: A Natural Experiment"

_ijerph, 2022, doi:10.3390/ijerph19052663_

Round 1
Reviewer 1 Report
- Introduction
In this part, the authors presented only the most important information. It is noteworthy that this part is not too extensive and the authors refer to relevant articles that can supplement the information. It is worth mentioning why the authors became interested in Netball.
it is worth quoting: https://doi.org/10.3390/sports10010012
- Materials and Methods
This part is bright and well covered. I miss the information when the research was done.
Please provide Cronbach's alpha for the WHO-5 index (e.g. DOI: 10.2337/dc07-0447 )
- Discussion
How does the program affect not only participants but other villagers? Did the authors notice any changes?
Reviewer 2 Report
The authors have done good work, and such programs should undoubtedly be included in other rural areas worldwide.
Ln35,36 – “This is particularly the case in Samoa, where the majority of the population do not meet the physical activity recommendations (51.7%) and the prevalence of diabetes..” – the sentence should end with (51%). Example: “..physical activity recommendations (51.7%). The prevalence of diabetes..” – please correct accordingly.
Ln163 –comparison group – control group sounds more appropriate.
Reviewer 3 Report
The rationale of that Sport-for-development programs can improve physical and mental health is good, and results of natural experiments are more likely to be applied to the field than laboratory experimental studies, so I think it was a good attempt. However, in order to conclude as an experimental study, there is a big problem in making excuses due to the limitations of the study, so this must be supplemented. In order for this manuscript to be published in Q1 or Q2 journal, the following must be revised and supplemented.
- The biggest problem is the dropout rate. The high proportion of dropouts in the experimental group is a representative factor that threatens the internal validity of the experimental research’ results. Many of the groups among originally planned as an experimental group were eliminated, and authors explained it with several other reasons, but in fact, it is highly likely that the participants gave up because the program did not work. Or it was a difficult group to apply this program. Look, the difference in pre-test scores is probably because of that. That's why the experimental group and the control group are different groups in the first place, so more changes have occurred in the experimental group. For example, not only the difference in the pre-scores of the dependent variables, but also ages (younger) in experimental group may have affected the change in weight due to sports participation. There are often dropouts in the experimental group, but in this study, there are too many and the impact is also shown in tables.
- Since this study was conducted as a natural experiment and the comparison group was not well controlled, it seems that you did not state it as a control group, but in general, the comparison group means a group with different treatment. Even if some of the people in the comparison group played netball, shouldn't it be expressed as a control group, not a comparison group, if the program applied in this study was not applied to the comparison group?
- Please explain more specifically about measurement tools, especially WHO-5. You have to let the reader know that it is a reliable and valid tool.
- I doubt how well the measurement was controlled. This is because there have been noticeable changes in the dependent variables in the comparison group. The influence of pre-test on post-test also lowers the internal validity of the results of pre-post experimental studies. There was a positive change in post-score only in self-reported tests, not in weight. Somethings may have affected the participant's responses in the post-test, like social desirability.
- Most psychiatrists or psychologists would disagree with the title of mental health based on what was measured whit WHO-5 well-being index. .
Round 2
Reviewer 3 Report
Thank you for revising your manuscript. It was supplemented based on what was commented in the first review, and because the subject of the study is interesting to the reader and has a good rationale, I believe that it is enough to be published in this journal if you describe the limitations of field experimental study properly.
Author Response
Thank you for your additional comments on our paper.
We have added further content to limitations section of the discussion to more explicitly address the concerns raised (Ln 347-351).
We hope that the paper now clearly addresses all of the reviewers methodological concerns and the related strengths / limitations of the approach we took in completing this study.